# HPV induced R-loop formation represses innate immune gene expression while activating DNA damage repair pathways

**Conor W. Templeton, Laimonis A. Laimins** [ORCID] *

Department of Microbiology-Immunology, Northwestern University Feinberg School of Medicine, Chicago, Illinois, United States of America

* l-laimins@northwestern.edu

## Abstract

R-loops are trimeric nucleic acid structures that form when an RNA molecule hybridizes with its complementary DNA strand, displacing the opposite strand. These structures regulate transcription as well as replication, but aberrant R-loops can form, leading to DNA breaks and genomic instability if unresolved. R-loop levels are elevated in many cancers as well as cells that maintain high-risk human papillomaviruses. We investigated how the distribution as well as function of R-loops changed between normal keratinocytes and HPV positive cells derived from a precancerous lesion of the cervix (CIN I). The levels of R-loops associated with cellular genes were found to be up to 10-fold higher in HPV positive cells than in normal keratinocytes while increases at ALU1 elements increased by up to 500-fold. The presence of enhanced R-loops resulted in altered levels of gene transcription, with equal numbers increased as decreased. While no uniform global effects on transcription due to the enhanced levels of R-loops were detected, genes in several pathways were coordinately increased or decreased in expression only in the HPV positive cells. This included the down-regulation of genes in the innate immune pathway, such as DDX58, IL-6, STAT1, IFN-β, and NLRP3. All differentially expressed innate immune genes dependent on R-loops were also associated with H3K36me3 modified histones. Genes that were upregulated by the presence of R-loops in HPV positive cells included those in the DNA damage repair such as ATM, ATRX, and members of the Fanconi Anemia pathway. These genes exhibited a linkage between R-loops and H3K36me3 as well as γH2AX histone marks only in HPV positive cells. These studies identify a potential link in HPV positive cells between DNA damage repair as well as innate immune regulatory pathways with R-loops and γH2AX/H3K36me3 histone marks that may contribute to regulating important functions for HPV pathogenesis.

## Author summary

R-loops are trimeric RNA: DNA hybrids that regulate transcription but can also lead to DNA breaks. R-loop levels are elevated in many cancers as well in precancerous cells containing high-risk HPVs. How the distribution and function of R-loops changes due to the

**Data Availability Statement:** All relevant data are in the manuscript and its Supporting information files. All raw datasets generated in this report have

been submitted to the NCBI GEO database and are public. These datasets can be found at GSE240391, GSE274109, GSE274120, GSE274122, GSE274119, and GSE274005.

**Funding:** This work was supported by grants to L. A.L. from the National Cancer Institute RO1CA142861 and RO1 CA59655 along with R21AI180285 from National Institute of Allergy and Infectious Diseases. L.A.L. received salary support from all three grants. C.W.T. salary was supported by the RO1CA142861, R21AI180285, and 5T32AR060710-10 from the National Institute of Arthritis and Musculoskeletal and Skin Diseases. The funders had no role in study design, data collection and analysis, decision to publish, or preparation of the manuscript.

**Competing interests:** The authors have declared that no competing interests exist.

presence of viral genomes was examined by comparing cells derived from an HPV 31 positive precancer (CIN 612) to normal keratinocytes. Up to 10-fold higher R-loop levels were detected on cellular genes in CIN 612 cells, with over 500-fold increases at ALU elements. Analysis of the changes in R-loop landscape between normal and HPV positive cells identified a functional association between R-loop formation, gene expression, and chromatin states. R-loop formation in HPV postive cells was equally associated with negatively and positively regulated gene expression; however, coordinated expression of genes in specific pathways by R-loops was observed. This included repression of genes in the innate immune surveillance pathway, while those regulating DNA damage repair and metabolism were activated.

## Introduction

R-loops are trimeric nucleic acid structures that are formed when an RNA strand hybridizes with its complementary DNA and displaces the opposite strand [1–5]. These structures are long-lived and regulate normal transcription as well as replication. Aberrant R-loops can form, and failure to efficiently resolve these structures leads to transcription/replication conflicts, resulting in DNA break formation [6–11]. High levels of R-loops have been detected in cell lines derived from precancerous lesions that maintain high risk human papillomaviruses (HPVs) [12–16]. Furthermore, human cancers themselves contain high levels of R-loops, which suggests they contribute to progression [17–20]. Few studies have, however, examined how R-loop distributions and functions differ between cells, such as those that maintain human papillomaviruses (HPVs) and normal cells. Our studies investigated how the landscape of R-loop distributions and functions change between cells that maintain high risk HPV-31 and normal keratinocytes.

HPVs are the etiological agents of cervical cancer and are responsible for ~5% of all human cancers. Cancers and precancers induced by infection with high-risk HPVs provide an excellent model for studying factors influencing progression [21–23]. Cervical lesions caused by high-risk HPVs are characterized as cervical intraepithelial neoplasia grades I to III (CINI-CINIII), and these precede the development of frank cervical cancer [24–26]. Characterization of lesions as CIN I to CINII is made according to the degrees to which epithelia are altered. In precancerous CIN I lesions, HPV genomes are maintained as extrachromosomal elements or episomes that replicate coordinately with cellular replication, while productive viral replication or amplification is restricted to differentiated suprabasal cells [27–29]. CIN 612 is an immortal cell line that was derived from a CINI cervical biopsy and stably maintains high-risk HPV 31 genomes as episomes [30]. Transfection of normal human keratinocytes with cloned HPV sequences leads to their immortalization and stable maintenance of viral episomes [31]. These cell lines are similar to those derived from CIN I lesions and demonstrate that viral genomes are responsible for changes indicative of precancerous lesions. Previous studies demonstrated the presence of high levels of R-loops in CIN612 cells in comparison to normal human keratinocytes (HFKs) [20,32]. Furthermore, R-loops formed on HPV genomes as well as cellular sites, and these high levels were found to be critical for viral transcription as well as replication. Elevated levels of R-loops have also been detected by immunofluorescence and immunohistochemistry analyses of biopsies from HPV positive cervical cancers [20,33]. In this study, we examined how the landscape of R-loops on cellular sites changes between HPV positive cells and normal keratinocytes as well as whether these alterations have functional consequences on cellular gene expression and HPV pathogenesis.

## Results

To investigate how the distributions and functions of R-loops change due to the presence of high-risk HPV genomes, we examined cells derived from a biopsy of an HPV 31 positive precancerous cervical lesion (CIN 612) and compared effects in normal keratinocytes (HFKs). Included in this initial analysis was the HFK-31 cell line that was generated by transfection of HFKs with cloned HPV 31 sequences and maintains viral sequences as episomes. Both HPV positive cell lines have been shown to exhibit similar histological changes in organotypic raft cultures consistent with CIN I lesions in vivo [31,34]. The levels of total R-loops in these cells were measured by dot blot assays that utilize the S9.6 antibody, which is specific for R-loops (Fig 1A). This analysis demonstrated that total R-loop levels in both HPV positive cell lines were significantly increased compared to normal keratinocytes. RNase H treatment abrogated these signals, demonstrating the assay was specific for R-loop formation. R-loops often form at promoter as well as transcription termination sequences, and the examination of these regions on a series of representative genes by DRIP-qPCR demonstrated substantially increased levels in HFK-31 and CIN 612 cells (Fig 1B). In this analysis, the levels of R-loops at previously reported sites in MYADM, RPL13a, SLC35B2, and LGAL2 were found to be increased on average by 5-to-10-fold relative to levels detected at the same sites in normal keratinocytes [12,16,35–37]. In addition, sites with low or negligible levels of R-loops associated with genes such as EGR1 and SNRPN in normal keratinocytes showed minimal increases in CIN 612 cells (Fig 1B). Increases of over 500-fold in R-loop levels were also detected in association with ALU elements, which may account for the higher total levels seen by dot blot analysis (Fig 1C). High levels of R-loops were also detected on HPV genomes at the early promoter and termination sites but not at coding sequences or the late poly A site (Fig 1C). These observations indicate there are substantial increases in R-loop levels in cells derived from HPV positive precancers or generated by transfection in comparison to normal keratinocytes and are consistent with previous reports [20].

DRIP-sequencing (DRIP-seq) was next performed to investigate how the distributions of R-loops varied between HPV positive cells and normal keratinocytes. This method allows for an unbiased approach to identify where R-loops are present within cells utilizing immunoprecipitations with the S9.6 monoclonal antibody followed by NEXTGen sequencing [38]. We focused this analysis on CIN612 cells in comparison to HFKs. Metaplot analysis of R-loop distribution of 2kb upstream and downstream of coding sequences in normal keratinocytes (HFK) and CIN 612 cells demonstrated that R-loop reads in both cell types peak near the transcription start site (TSS), at the transcription end site (TES), and about 1–1.5kb downstream of the TES. This distribution is similar to the profile published by Promonet et al. [12]. For our downstream analyses, R-loops within these regions were associated with a gene's coding region and referred to as genic R-loops. Importantly, the overall R-loop distributions at TSS and TES sites are similar in CIN 612 cells (Fig 1D, left) as well as in normal keratinocytes (Fig 1D, right). The primary difference between the two cell types was that the signal in the CIN 612 cells was significantly higher than in the normal keratinocytes. Heatmap analysis demonstrated that genes with high levels of R-loops at the TSS also had high levels of R-loops at the TES (Fig 1D, bottom). Only a minority of genes exhibited distinct patterns of R-loop formation in CIN 612 compared to normal keratinocytes. Examples of R-loops distributions showing the IP/input enrichment on four different genes are shown in Fig 1E. RPL13a, MYADM, and LGALS2 all contained significantly higher R-loops levels over the input background control in the precancerous CIN 612 cells than in normal keratinocytes, consistent with our DRIP-qPCR analysis (Fig 1E to 1B). ZNF554 is a gene not associated with R-loop formation, and it exhibited minimal R-loop levels over the input background in both cell lines examined.

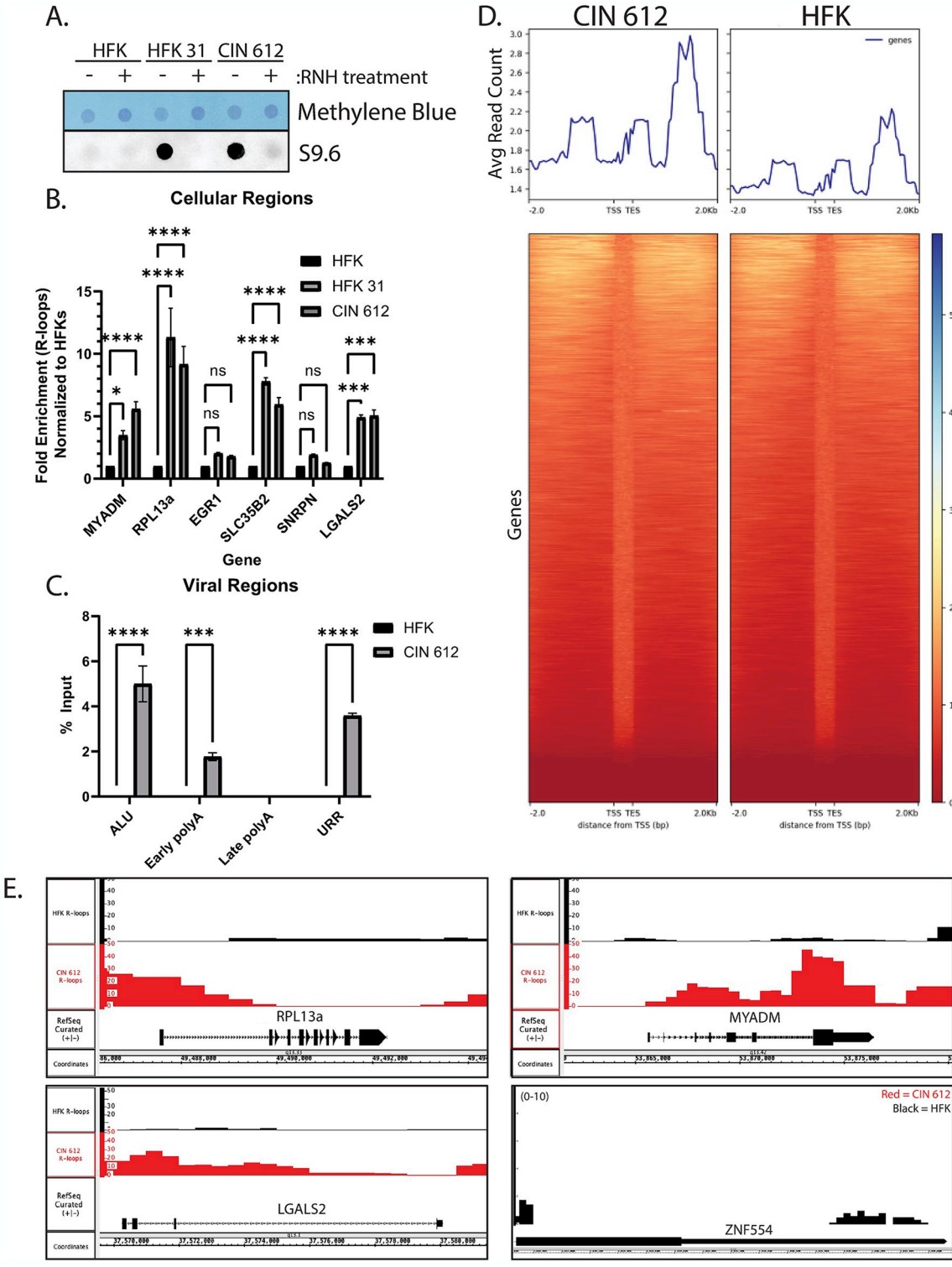

**Fig 1. Distribution of R-loops in HPV positive cells compared to normal keratinocytes.** (A) S9.6 dot blot analysis from whole cell nucleic acid extracts of normal keratinocytes (HFKs), transfected, HPV positive keratinocytes (HFK-31), and HPV positive keratinocytes derived from a cervical CIN 1 lesion (CIN 612). Total nucleic acid levels were measured via methylene blue staining (top), and the specificity of the S9.6 monoclonal antibody was assessed through RNase H treatment. (B) DNA:RNA immunoprecipitation assays (DRIP) were performed on HFKs, HFK 31, and CIN 612 cells for six representative cellular sites, and immunoprecipitated chromatin was analyzed by quantitative PCR (qPCR). Primers mapped to EGR1, RPL13a, SLC35B2, and LGALS2

were used as positive controls for regions previously characterized to contain R-loops; SNRPN was used as a negative control, while MYADM has variable reports of its association with R-loops [12,16,35–37]. Fold enrichment for each primer set over HFK$_{S9.6}$ is plotted: $(S9.6_x/IgG_x)/ (S9.6_{HFK}/IgG_{HFK})$ where x is Ct values from either HFK 31 or CIN 612 cells. The error bars represent the standard error of the mean (n = 3, ns, not significant; p<0.05, *; p<0.001, ***; p<0.0001, ****). (C) DRIP-qPCR of three regions on the HPV 31 genome in comparison to ALU repetitive cellular elements. DRIP-qPCR was performed on HFKs, HFK 31, and CIN 612 cells using primers mapping to ALU elements and viral genomic elements (early polyA site, upstream regulatory region (URR), and the late polyA site). Percentage input was plotted: Input % = $100/2^{(\Delta Ct \; [normalized \; to \; input \; control])}$. The error bars represent the standard error of the mean (n = 3, p<0.001, ***; p<0.00001, ****). (D) Metaplot distribution of S9.6 signal (IP—input) through genic regions, including 2kb flanking upstream or downstream (n = 2, top). Heat map of S9.6 intensity through genic and 2kb flanking regions (n = 2, bottom). (E) Depth graphs of input normalized S9.6 reads through MYADM, LGALS2, RPL13a, and ZNF554 in normal keratinocytes (black) and CIN 612 cells (red).

S1 Fig shows similar analyses for 2 additional genes (DNA lig IV and CALML5). This overview indicates that enhanced levels of R-loops are present in CIN 612 cells compared to normal keratinocytes, and these form at similar, though not necessarily identical, regions in the proximal 2kb upstream and downstream regions.

## Identification of genomic regions enriched with R-loops

Further analysis of the DRIP-seq data was then used to provide an overall picture of which sites were associated with enhanced R-loop formation in CIN 612 cells as compared to normal keratinocytes. Peak calling analysis demonstrated that R-loops were significantly enriched over background at over 90,000 sites in HPV positive cells and at approximately 40,000 sites in normal keratinocytes. About 30,000 sites were shared between both cell lines, leaving over 60,000 unique R-loop sites in CIN 612 cells and approximately 9,500 unique sites in normal keratinocytes (Fig 2A). MA plot analysis of the ~30,000 common sites indicated significantly higher R-loop levels at those sites in CIN 612 cells compared to normal keratinocytes, with some R-loop sites being enriched by almost ~1000 fold in the HPV positive cells (Fig 2B). Although there are significant differences in the numbers of R-loop reads between CIN 612 cells and normal keratinocytes, peak distribution over promoter, exonic, and downstream sequences were very similar, with approximately 60% of the R-loop peaks detected in genic regions in both cell types (Fig 2C).

Since the levels of R-loops were substantially increased in HPV positive cells relative to normal keratinocytes, it was possible they were linked to specific genes or pathways. For this analysis, we first examined genes associated with R-loops in both normal keratinocytes and CIN 612 cells, focusing on proximal promoter, gene body, and terminator regions. Pathway analysis on the R-loop containing genes was then performed using ShinyGO 0.80 [39]. The KEGG pathways associated with high levels of R-loops in CIN 612 cells included those involving cancer progression (pathways in cancer, proteoglycans in cancer, and transcriptional misregulation in cancer) and DNA virus infection, many of which were not found to be enhanced in normal keratinocytes (Fig 2D, top). In contrast, pathway analysis of genes containing R-loops in normal keratinocytes identified prominent pathways involved in clathrin binding, lipid binding, and kinase activity (Fig 2D, bottom). These analyses indicated that the increased formation of R-loops in CIN 612 cells is localized to genes in pathways that are distinct from those seen in normal keratinocytes. It was next important to determine if this increased R-loop association correlated with increased transcription.

## HPV positive cells have similar numbers of genes upregulated and downregulated, despite high R-loop levels

In order to determine if there was a correlation between high levels of R-loops and increased transcription of the associated genes, RNA sequencing was performed on CIN 612 [20] and

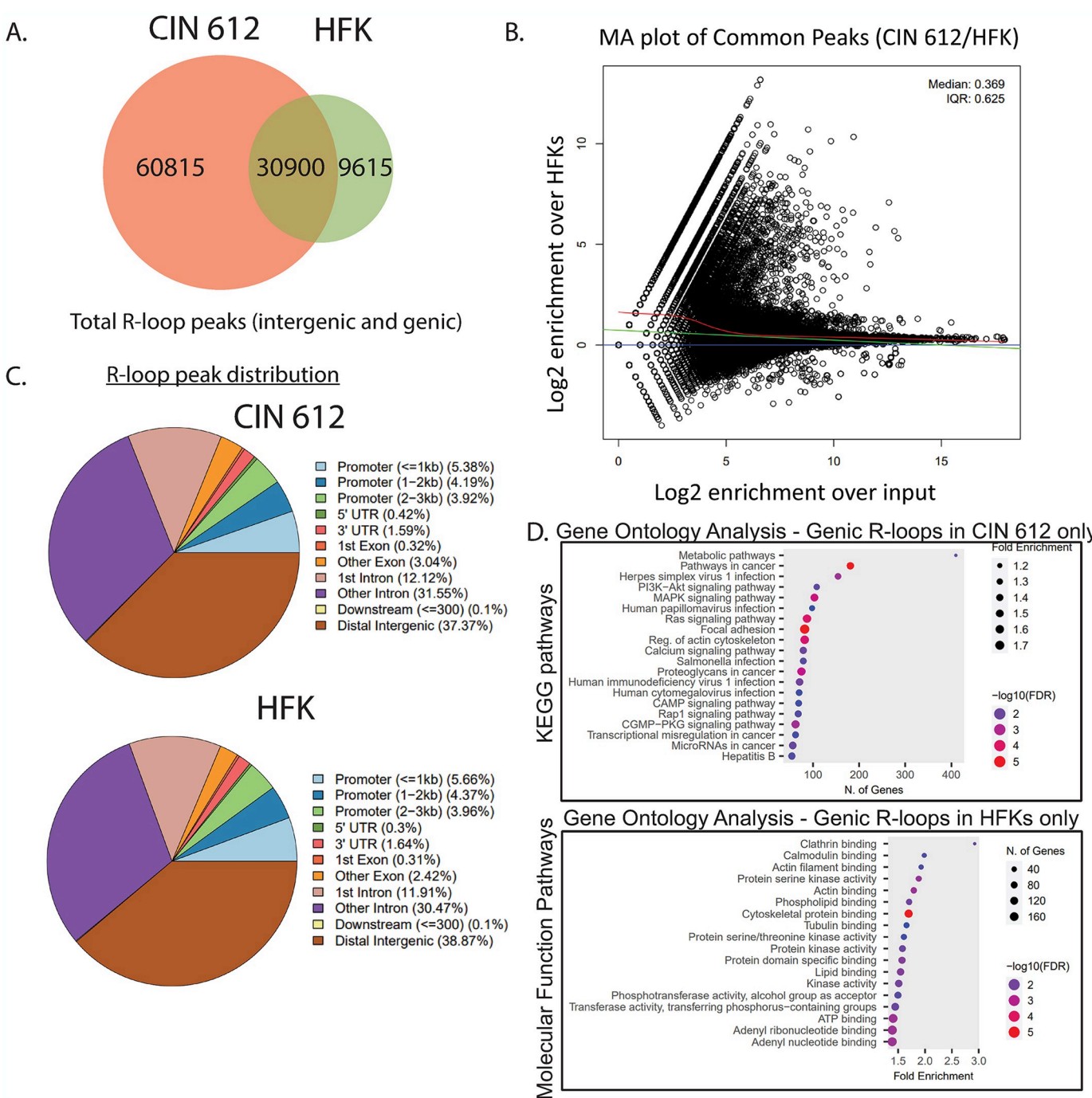

**Fig 2. R-loops form preferentially on genes in pathways responsible for the cancer progression and viral pathogenesis in CIN 612 cells.** (A) Venn diagram of the genomic regions containing R-loop peaks (MACS) overlapping between normal keratinocytes and HPV positive cells (CIN612) (n = 2, a representative image is shown). Total R-loops in CIN 612 (pink) and HFK (red) with common sites (brown). (B) MA plot analysis of the common R-loop containing genes between CIN 612 cells and normal keratinocytes. Log$_2$ enrichment of R-loop levels over the matched input control is plotted on the x-axis, and Log$_2$ enrichment of R-loop levels in genes present within CIN 612 cells over HFKs is plotted on the y-axis. (C) Distribution of R-loop peaks relative to genomic locations in HFKs and CIN 612 cells. CHIPSEEKER was used to analyze the location of R-loop reads within each sample. (D) HOMER was used to identify the location of where R-loop peaks occurred within HFK and CIN 612 cells. Intergenic R-loops were filtered out, leaving R-loops that fell within introns, exons, TES, TSS, 3'UTR, and 5'UTR. Common genes found in both HFKs and CIN 612 cells were also filtered out. Pathway analysis was then performed on the genes to which these R-loops were assigned to either the CIN 612 cells or the HFKs using Shiny GO 0.80. KEGG pathways or molecular function analyses are shown.

normal keratinocytes. This analysis demonstrated that approximately 20% (~4,500) of the genes analyzed were differentially expressed compared to normal cells. Interestingly, these genes were divided almost equally between those upregulated (2,207) and those downregulated (2,280) (Fig 3A). A similar distribution in the fold changes of differentially expressed genes was seen by MA plot analysis (Fig 3B). We then performed pathway analysis of the differentially expressed genes in CIN 612 cells compared to normal keratinocytes to determine which pathways were altered (Fig 3C). Consistent with previous findings, genes in pathways involved in DNA repair, DNA replication, and cell cycle were upregulated, while those in pathways important for epithelial differentiation and epidermal development were downregulated [40,41]. In addition, pathways involved in the immune response and interferon signaling were downregulated in CIN 612 cells.

DRIP-seq analyses were then used to investigate if altered expression levels were linked to increased R-loop levels. This analysis demonstrated an approximately 2-fold higher level of total transcripts associated with genes that were linked to R-loops compared to those without R-loops (Fig 3D). Importantly, approximately 30% of the differentially expressed genes were found to be associated with R-loops only in CIN 612 cells and absent in normal keratinocytes (Fig 3E). These differentially regulated genes were similarly distributed between those downregulated (672) and those upregulated (594). While there was not a consistent global increase or decrease in the expression of genes that were uniquely associated with R-loops in CIN 612 cells, KEGG analysis identified genes in specific pathways that were coordinately regulated (Fig 3F). The most prominent pathways associated with enhanced R-loop levels were involved in replication and DNA metabolism. At the same time, genes associated with the innate immune surveillance pathway, including IL1B, STAT1, and MYD88, were linked to enhanced R-loop levels but exhibited decreased expression in CIN 612 cells relative to normal keratinocytes. Furthermore, no correlation was found between the enrichment of R-loops and their corresponding mRNA levels with respect to whether these structures formed at either TSS, TES, or intronic locations (S2 Fig).

The above studies indicated there was a correlation between the presence of unique R-loops in CIN 612 cells and altered expression of genes in specific pathways. It was next important to determine if their expression was functionally dependent upon enhanced levels of R-loops. For this analysis, we utilized CIN 612 cells that were generated to overexpress RNase H1, an R-loop processing enzyme, by transfection of CMV-directed tagged expression vector followed by selecting stable cell lines [42]. Overexpression of RNAse H1 has been characterized as the "gold standard method" for reducing R-loop levels [43]. When RNase H1 was overexpressed in CIN 612 cells, viral early gene expression and episome levels were reduced by ~70% and 50%, respectively. As HPV 31 E6 was shown to induce R-loop formation in HFKs, this could further enhance these reductions [20]. RNA-seq analysis was previously performed on these cells and compared to that seen with parental CIN 612 cells [20]. The cells overexpressing RNase H1 exhibited substantially reduced levels of R-loops compared to the parental line. Around 12% of all genes (FKPM > 0) were differentially expressed within CIN 612 cells overexpressing RNase H1 compared to the scramble control cells (S3A Fig). Furthermore, 833 of the genes were associated with R-loops only in CIN 612 cells, further supporting a direct functional relationship between R-loop formation and gene expression (S3B Fig).

Pathway analysis of genes whose expression was dependent upon R-loops present only in CIN 612 cells were linked to innate immune surveillance, including interferon-alpha and interferon-gamma responses, complement signaling, and inflammation (Fig 4A). Genes in these innate immune surveillance pathways that are repressed by R-loops in CIN 612 cells include STAT1 (8-fold), NLRP3 (14-fold), JAK2 (4-fold), AIM2 (34-fold), RIG-I (22-fold), IFNB (greater than 50-fold), and IL6 (190-fold). In contrast, TRIM 14 and STING are only

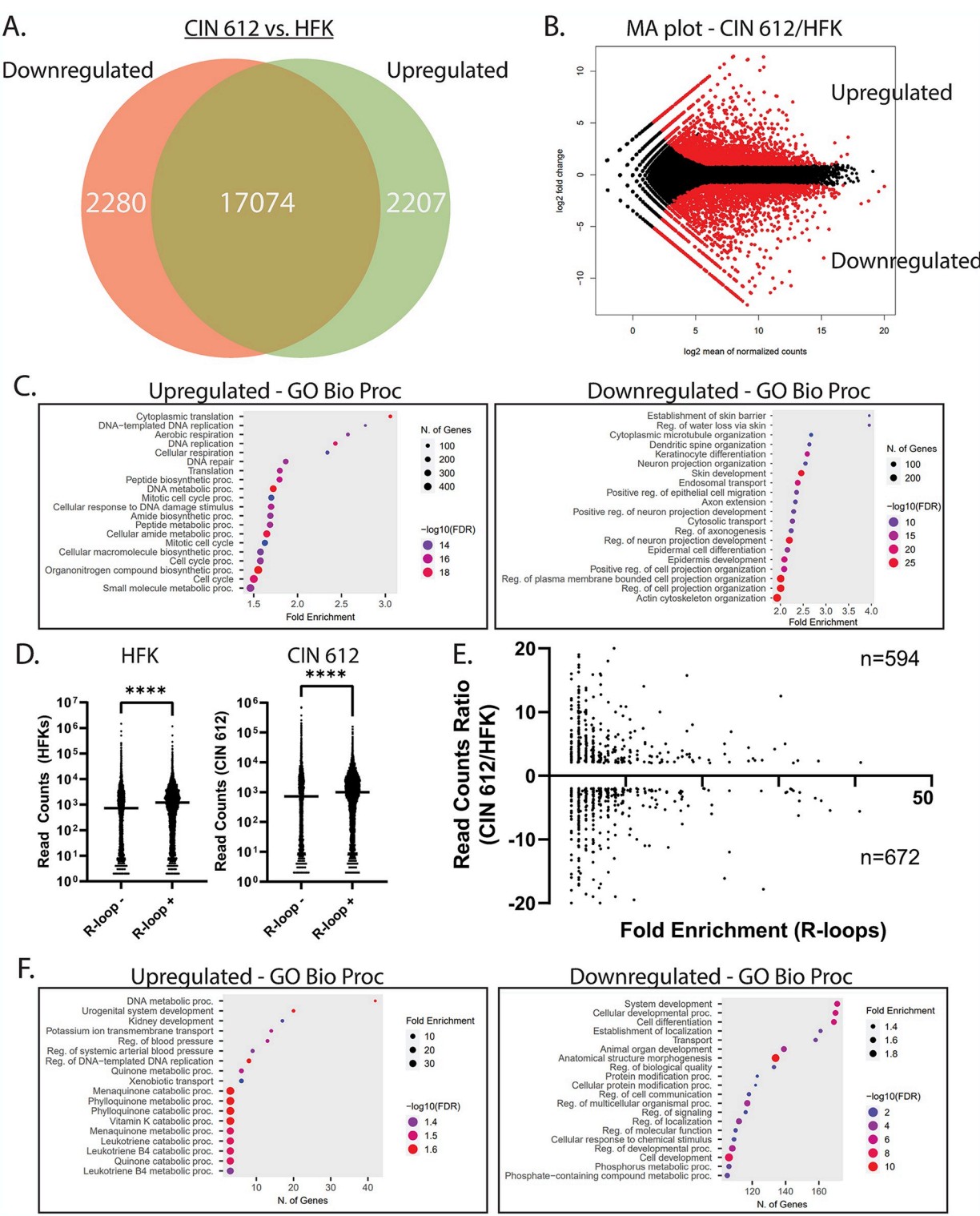

**Fig 3. Formation of R-loops at unique sites in HPV positive, CIN 612 cells correlates with differential gene expression.** (A) Venn diagram of the differentially expressed genes in precancerous CIN 612 cells compared to normal keratinocytes (n = 2, RPKM > 0) Genes upregulated in CIN 612 (green), upregulated in HFKs (red), and those with no difference (<1 Log$_2$ FC, brown). (B) MA plot analysis of the differentially expressed genes exhibiting a similar distribution of downregulation and upregulation. (C) The differentially expressed genes were divided between those upregulated and downregulated in CIN 612 cells compared to HFKs. Pathway analysis of the biological processes of these genes was performed using Shiny GO 0.80. (D) mRNA levels of genes R-loop positive or negative in HFKs and CIN 612 cells. The line represents the mean (p<0.0001;

****). (E) Around 25% of all differentially expressed genes are associated with R-loops only in precancerous CIN 612 cells. R-loop peaks in the genic or 2kb flanking regions of genes in HFKs or those that were common to both HFKs and CIN 612 cells were filtered out. The remaining genes were screened against the differentially expressed genes in CIN 612 cells compared to HFKs. Fold enrichment of S9.6 reads over input is plotted on the X-axis, while fold change of mRNA levels in CIN 612 cells compared to HFKs is plotted on the Y-axis (left). (F) Pathway analysis of the R-loop containing genes upregulated (left) and downregulated (right) in CIN 612 cells.

modestly repressed (Fig 4B). The expression of some genes, such as IFN β and RIG-I, increased in response to reductions in levels of R-loops despite not being physically linked with these structures. This likely indicates that R-loops target their upstream regulators, so it is likely that the effects on IFN β and RIG-I are indirect.

## Histone modifications are differentially deposited on host chromatin within HPV positive cells

The linkage of enhanced levels of R-loops with coordinated expression of genes in multiple specific pathways suggested that additional factors act to facilitate this specificity. One way R-loops could coordinate the expression of genes in distinct pathways might be through association with different sets of modified histones that configure chromatin around these structures [44]. In addition to histones linked to chromatin states, R-loops are also associated with the modified histone, γH2AX, which is coupled with DNA break formation and may be linked to gene expression [12,45,46]. Therefore, we investigated whether there are associations between specific sets of histones and R-loop-directed gene expression that vary between normal and HPV-positive cells.

For this analysis, chromatin immunoprecipitation was performed on CIN 612 cells and normal keratinocytes for three histone marks: H3K36me3, H3K9me3, and γH2AX. H3K36me3 is typically associated with transcription, while H3K9me3 marks areas of heterochromatin [47–50]. We performed peak calling algorithms on each of the modified histones pulldown experiments using the MACS peak calling algorithm. We controlled for off-target pulldowns by background subtracting respective input control samples isolated from each of the cell lines (HFK and CIN 612 cells). The called peaks were then assigned a relative genomic location using HOMER. Peak calling analysis for these histone marks focused on the regions 2kb upstream, 2kb downstream, or in the gene body in both CIN 612 and normal keratinocytes. This analysis identified an overlap of these histones with unique and common sets of genes associated with R-loops. Overall, H3K36me3 marks were approximately 4-fold more prevalent in CIN 612 cells than in normal cells (S4 Fig). In contrast, the opposite was found for H3K9me3 marks which were reduced in CIN 612 cells relative to normal keratinocytes (S4 Fig).

It was next important to determine whether the presence of H3K36me3 and H3K9me3 formation correlated with the enhanced formation of R-loops and transcription in HPV positive cells by comparing ChIP-seq analysis for these histones to DRIP-seq and RNA-seq data, respectively. RNA-seq analysis of genes containing H3K36me3 marks identified an over 2-fold enrichment in mRNA levels of these genes over those that did not contain H3K36me3 in both normal and CIN 612 cells (Fig 5A, left). In contrast, the H3K9me3 mark on genes that were linked to R-loops in normal keratinocytes correlated with a modest 0.33-fold increase in mRNA levells, with no significant difference seen in the CIN 612 cells (Fig 5A, right). Comparing the ChIP-seq analyses with DRIP-seq, over 50% of the R-loop containing genes in CIN 612 cells were also positive for H3K36me3 (Fig 5B, bottom left), while less than 8% were associated in normal keratinocytes (Fig 5B, top left). In contrast, less than 2% of genes with R-loops in CIN 612 cells were H3K9me3 positive as compared to over 10% in normal keratinocytes

**A.**

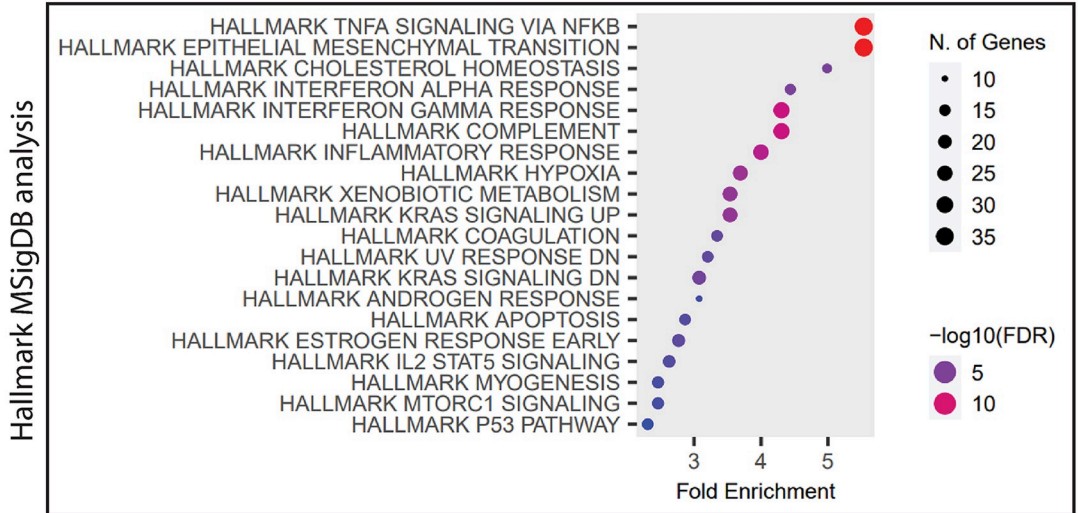

**B.**

### R-loop regulated expression of immune response genes in precancerous cells

| | Fold Change (Normalized to HFKs) | | | Presence of unique genic mark in CIN 612 cells | | |
|---|---|---|---|---|---|---|
| **Gene** | **HFK** | **Scrm CIN 612** | **oeRNaseH1 CIN 612** | **R-loops** | **H3K36me3** | **γH2AX** |
| NLRP3 | 1 | 0.696 | 9.174 | + | + | - |
| MYD88 | 1 | 0.256 | 1.028 | + | + | - |
| JAK2 | 1 | 0.807 | 2.957 | + | + | + |
| AIM2 | 1 | 3.333 | 114.667 | + | + | - |
| IL1B | 1 | 0.098 | 0.406 | + | + | - |
| STAT1 | 1 | 0.338 | 2.493 | + | + | - |
| DDX58 | 1 | 0.421 | 11.15 | + | + | - |
| IFNK | 1 | 0.001 | 0.012 | - | + | - |
| IFNB1 | 1 | 0 | 59.333 | - | - | - |
| IL6 | 1 | 3 | 569.5 | + | + | + |
| TLR3 | 1 | 1.202 | 3.783 | - | - | - |
| TRIM14 | 1 | 0.295 | 0.669 | + | + | + |
| TRIM25 | 1 | 0.505 | 1.708 | - | + | + |
| TRIM38 | 1 | 0.425 | 1.43 | - | + | - |
| MDA5 | 1 | 0.63 | 7.16 | + | + | - |
| cGAS | 1 | 3.87 | 6.22 | + | + | + |
| IRF7 | 1 | 0.91 | 10.95 | + | + | - |
| LGP2 | 1 | 1.03 | 12.12 | + | - | - |
| Caspase 7 | 1 | 1.33 | 4.01 | + | + | - |

**Fig 4. Reduction of R-loops through RNase H1 overexpression in CIN 612 cells identifies genes that are functionally dependent on their formation.** (A) Major pathways whose expression is dependent on R-loops present only in CIN 612 cells as determined by RNase H1 overexpression. Pathway analysis was performed using the Hallmark database in Shiny GO 0.80 of R-loop regulated genes that contain R-loops unique to the CIN 612 cells. (B) Innate immune response genes whose expression is regulated by R-loops that are present only in CIN 612 cells. Fold changes are normalized to the mRNA counts in normal keratinocytes (n = 2). Linkages to H3K36me3 and γH2AX histones are shown on the right.

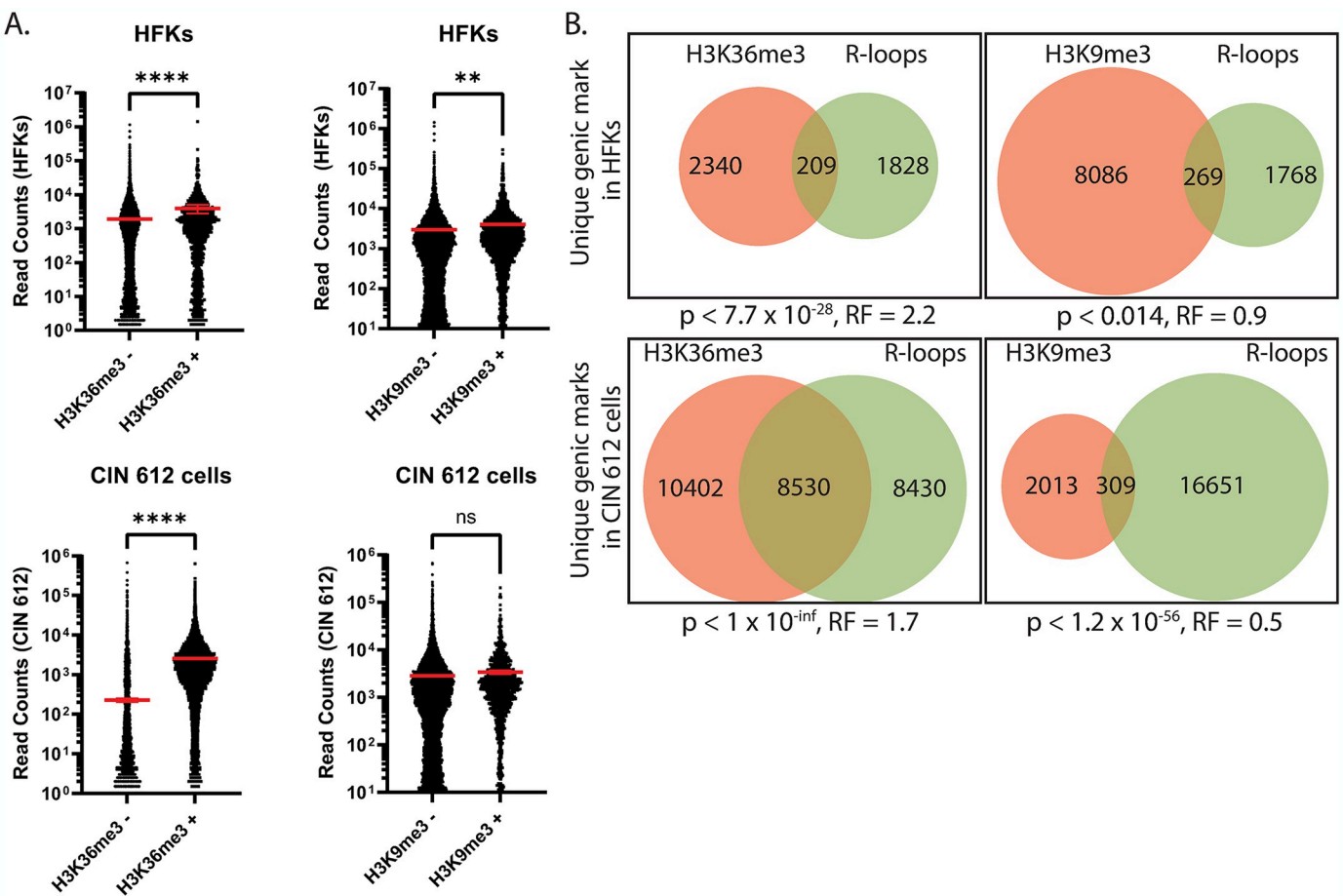

**Fig 5. Modified histones are differentially associated with actively transcribed genes and R-loops in HPV positive cells.** (A) Total mRNA levels of genes that are H3K36me3 positive or negative (left) as well as H3K9me3 positive or negative (right) in HFKs (top panels) and CIN 612 cells (bottom two panels). The red line represents the mean. The error bars are SEM (ns, not significant; p<0.0001, ****). (B) Venn diagrams of the genomic regions containing H3K36me3 (left) or H3K9me3 (right) and R-loop peaks (MACS) overlapping between the HPV negative (HFKs) and positive (CIN612) cells (n = 2, a representative image is shown). Genes containing H3K36me3 or H3K9me3 peaks (red), R-loop peaks (green), and both (brown). P-values represent that the overlap between genic marks is <u>above</u> or below that expected from random distribution. RF represents representation factors. RF values greater than 1 suggest more overlap than expected from random distribution, while RF values less than 1 imply less overlap.

(Fig 5B, right). Gene ontology analysis of the H3K36me3 and R-loop positive genes in CIN 612 cells identified pathways associated with cell cycle progression and innate immune surveillance (S5 Fig, top left). Genes in the immune response pathways identified above exhibited a strong linkage between the presence of both R-loops and H3K36me3 (Fig 4B). In contrast, in normal keratinocytes, genes associated with H3K36me3 and R-loops were found in distinctly different pathways that regulate membrane potential, protein localization, and neurogenesis (S5 Fig, top right).

HPV positive cells contain high levels of modified H2AX ($\gamma$H2AX) and DNA breaks, which results in the constitutive activation of DNA damage repair pathways [28,51]. Consistent with these findings, peak calling analysis identified ~4-fold more $\gamma$H2AX marks (21,941 vs. 4,870) in the CIN 612 cells than in the normal cells (S6 Fig). In addition, the profile of averaged $\gamma$H2AX reads across genic regions of precancerous CIN612 cells differed substantially from that seen in the normal keratinocytes. While normal keratinocytes exhibited no significant increases in $\gamma$H2AX reads across genic regions, $\gamma$H2AX reads in HPV positive cells increased,

peaking at the TES (S6 Fig). Pathway enrichment analysis of genes containing γH2AX marks in the precancerous CIN 612 cells identified those responsible for cell cycle control, regulation of RNA biosynthetic processes, and transcription. This enrichment was only seen in HPV positive cells and not in normal keratinocytes. Interestingly, γH2AX was associated with genes with ~2-fold higher transcript levels than those without γH2AX (Fig 6A). Approximately 37% of all genic R-loops in CIN 612 cells were found to be associated with γH2AX in contrast to normal keratinocytes, where negligible levels were detected (Fig 6B). It was next important to determine whether H3K36me3 or H3K9me3 were preferentially associated with γH2AX and R-loops. Interestingly, about 25% of all R-loop-containing genes in CIN 612 cells were found to be positive for the combination of R-loops, H3K36me3, and γH2AX (Fig 6C). This level of correlation was not seen in normal cells nor with H3K9me3 and γH2AX. The R-loop associated genes which were H3K36me3 and γH2AX positive, are involved in pathways essential for viral replication, including DNA break repair and cell cycle control (Fig 6C, right).

The association of H3K36me3, γH2AX, and enhanced R-loops was particularly significant for genes in the DNA repair pathway. HPV proteins activate the ATM and ATR DNA repair pathways, which is critical for differentiation-dependent amplification. Our studies show that genes like ATM, ATRX, RAD51C, along with members of the Fanconi Anemia pathway (FANC-B, C, E, I, L, and M), and SETD2, the methyltransferase regulating H3K36me3, were all associated with the combination of H3K36me3, γH2AX and enhanced R-loops (Fig 6D). While a significant linkage was found between innate immune regulatory genes and the presence of both R-loops and H3K36me3, only a minimal association was found for the combination of H3K36me3, γH2AX, and enhanced R-loops. This suggests there may be a preferred linkage of H3K36me3, γH2AX, and enhanced R-loops with genes in the DNA damage repair pathway. These results indicate that the linkage between all three factors, γH2AX, H3K36me3, and R-loops, is critical for HPV pathogenesis and cancer progression.

## Discussion

The levels of R-loops are increased in many cancers, and how the distribution, as well as the function of these structures, change due to the presence of high-risk HPV genomes was examined by comparing cells derived from an HPV 31 positive precancerous lesion of the cervix (CIN I) to normal keratinocytes. The levels of R-loops were found to be enriched by ~5–10 fold on individual cellular genes in CIN 612 cells in comparison to normal keratinocytes. The largest enrichment of R-loops identified in HPV positive cells was, however, associated with repetitive ALU elements, which exhibited over a 500-fold increase compared to that seen in normal keratinocytes. While the levels of R-loops are significantly increased in HPV positive cells, the overall pattern of where R-loops form on cellular genes is very similar to that detected in normal keratinocytes, with peak levels located within 2kb upstream of start sites, within gene bodies, as well as 2kb downstream of termination sequences. Approximately one-third of the R-loops identified in CIN 612 cells are located at sites similar to those found in normal keratinocytes, while about two-thirds of the R-loops are associated with unique genes only in the HPV positive cells and not in normal keratinocytes. Interestingly, the expression of genes with R-loops associated only in CIN 612 cells is divided equally between those with increased or decreased transcript levels. While no global increase in expression is associated with enhanced R-loop levels, genes in specific pathways were found to be coordinately regulated. This includes pathways associated with DNA repair, DNA replication and cell cycle, whose expression is coordinately increased. Equally interesting is the identification of genes involved in innate immune surveillance and keratinocyte differentiation, which are suppressed. All these changes may contribute to progression from normal to precancerous states as well as for the

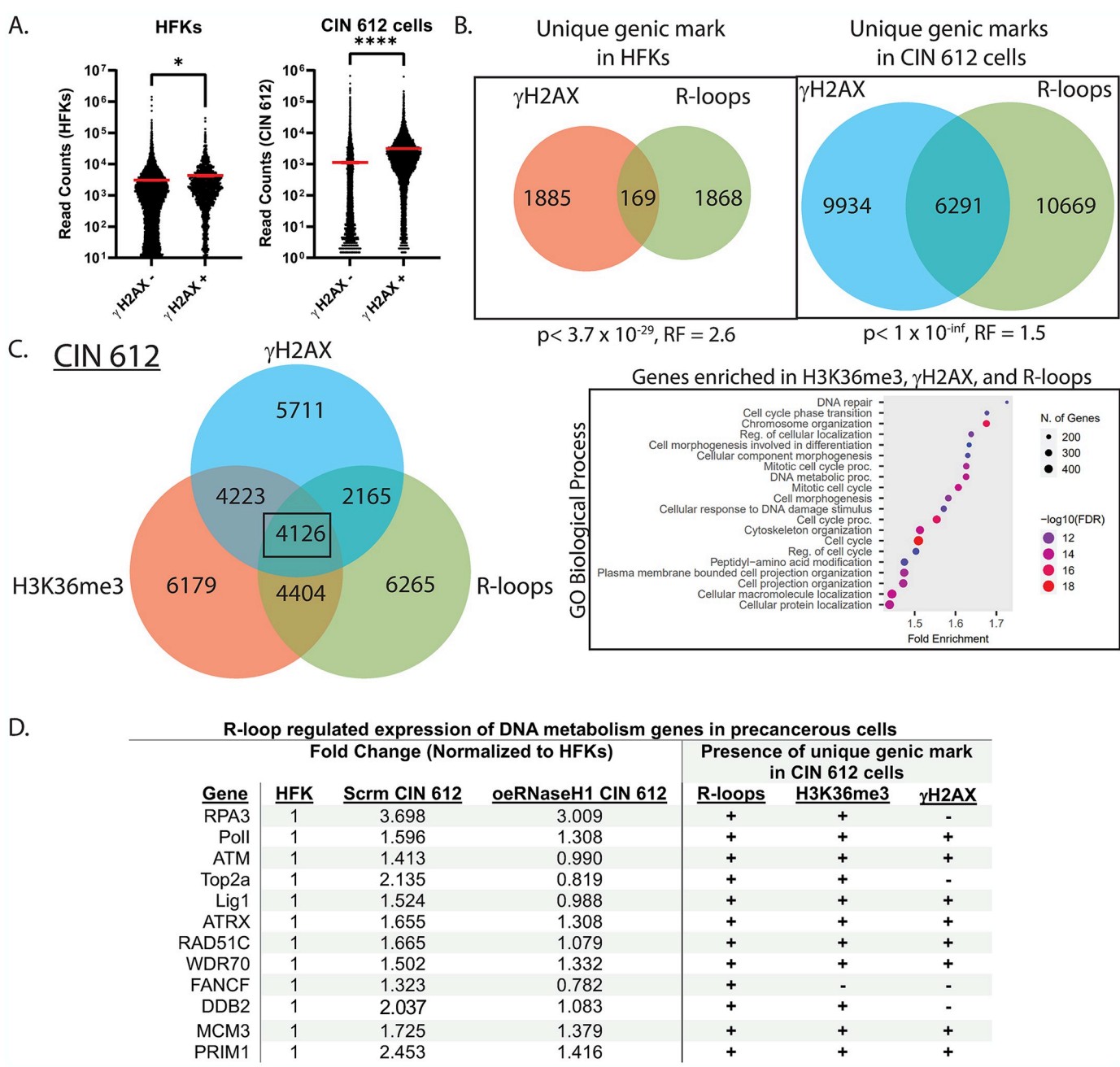

**Fig 6. γH2AX is associated with R-loop formation and H3K36me3 deposition in CIN 612 cells.** (A) mRNA levels of genes γH2AX that are negative or positive in HFKs (left) and CIN 612 (right) cells. The red line represents the mean. The error bars are SEM (p<0.05, *; p<0.0001, ****). (B) Venn diagrams of the genes containing γH2AX and R-loop peaks (MACS) overlapping between the HFKs (left) and CIN 612 (right) cells (n = 2). Genes containing γH2AX peaks (red), R-loop peaks (green), and both (brown). P-values represent that the overlap between genic marks is <u>above or below</u> that expected from random distribution. RF represents representation factors. RF values greater than 1 suggest more overlap than expected from random distribution, while RF values less than 1 imply less overlap. (C) Venn diagrams of the genes containing γH2AX, H3K36me3, and R-loops in CIN 612 cells (left) and the GO biological processes pathway analysis of these genes (right). (D) A table showing DNA repair and metabolism genes that are differentially expressed in CIN 612 cells with corresponding marks of R-loops, γH2AX, and H3K36me3, which are present on these genes only in the precancerous cells.

pathogenesis of high-risk HPVs, which are the etiological agents responsible for cervical intraepithelial neoplasia. This indicates that the directed formation of R-loops on specific groups of genes may provide an important function in the HPV life cycle.

The repression of genes in the innate immune surveillance pathway in the CIN 612 cells is particularly sensitive to enhanced levels of R-loops. In wild type CIN 612 cells, the expression of many innate immune regulatory genes is reduced by 2 to 5-fold from that seen in normal keratinocytes (Fig 4B). The stable overexpression of RNase H1 in CIN 612 cells resulted in increased expression of innate immune genes, including STAT1 (8-fold), NLRP3 (14-fold), AIM2 (34-fold), IL6 (190-fold), and IFNβ (over 50-fold), indicating their repression may be functionally linked to R-loop levels. Importantly, R-loops are only found to be associated with these cellular genes in the CIN 612 cells. Some of these increases exceed the amounts seen in normal keratinocytes, suggesting that multiple upstream regulators of these factors also depend on R-loop formation. Furthermore, the expression of several genes, such as RIG-I and TRIM 25, are increased by RNase H1 overexpression despite not being linked to the presence of R-loops. These are interferon stimulated genes, and the increases in expression are likely the result of the enhanced levels of IFNβ that are induced when R-loop levels are reduced [52–57]. Along with increased expression of innate immune genes, RNase H1 overexpression also reduces viral gene expression and episome levels by ~70% and 50%, respectively. Whether these reductions are due to increased expression of innate immune regulators or a direct effect due to loss of R-loops on viral episomes is unclear. In contrast to the repression of the innate immune regulatory pathway, genes in the DNA damage repair pathway are activated by enhanced levels of R-loops in CIN 612 cells. This includes genes such as ATM, Top2A, Lig1, and RAD51C. While the differences seen with the DNA damage repair genes are not as dramatic as seen with the innate immune regulated genes, the activation of the DNA damage repair pathway in HPV positive cells has been shown to be critical for viral pathogenesis and cancer progression.

Our observation that R-loops are found in association with specific sets of genes that are linked with both increased and decreased expression indicates that their formation is not merely an accidental byproduct of increased transcription but is instead the result of a directed process. One way that expression could be linked with enhanced levels of R-loops is through altered chromatin states associated with specific sets of modified histones. Previous studies have suggested an association of H3K36me3 and H3K9me3 with certain classes of R-loops, but only a limited correlation with altered expression has been described [2,58]. Our studies demonstrated that over half of R-loop associated genes in CIN I derived cells are associated with H3K36me3 marks, while only 8% are positive in normal keratinocytes. H3K36me3 has been linked with increased transcription; however, in our study, equal numbers of dually H3K36me3 and R-loop positive genes exhibit increased expression as decreased expression compared to normal keratinocytes [59–62]. This indicates that this histone mark is more likely associated with an accessible chromatin configuration rather than increased transcription alone. Both innate immune regulatory genes, as well as those in DNA damage repair, are linked with high levels of H3K36me3, and this is not seen in normal keratinocytes, demonstrating that this effect is specific to HPV positive cells. SETD2 is the methyltransferase that regulates the deposition of methyl groups on lysine 36 of histone 3 (H3K36me3), and its levels are increased in CIN 612 cells as well as other HPV positive cells [63–65]. Knockdown of SETD2 in HPV positive cells has been shown to lead to significant reductions in viral episomes, identifying it as an important regulator of viral persistence. While H3K36me3 has been identified as a mark of transcription elongation, recent studies have also linked it with DNA repair suggesting a potential link with genomic instability and DNA breaks [66–68]. A previous study linked cells with high R-loop levels to concomitant decreases in H3K9me3 levels,

consistent with our studies, as CIN 612 cells contained far fewer of these marks than normal keratinocytes [69]. In contrast, no strong linkage was found between H3K9me3 and R-loop regulated gene expression in CIN 612 cells. Only 2% of R-loop associated genes were also positive for H3K9me3 as compared to 10% in normal keratinocytes.

The failure to resolve R-loops leads to the formation of DNA breaks and genomic instability [70]. HPV positive precancers, as well as other cancers, exhibit high levels of DNA break formation as indicated by enhanced amounts of γH2AX, which is often used as a surrogate marker [71]. In CIN 612 cells, high levels of γH2AX are associated with increased levels of R-loops at genes whose expression is altered. Over one third of the genes associated with R-loops in the HPV positive cells were also positive for γH2AX. In addition, 67% of the genes positive for both γH2AX and R-loops were also linked to H3K36me3. No such associations are seen in normal keratinocytes. Approximately 700 of the genes that are differentially expressed in the CIN 612 cells are linked to the combined presence of γH2AX, H3K36me3, and R-loops. Genes whose expression is positively regulated by R-loop formation and associated with both γH2AX and H3K36me3 include ATM, ATRX, ATR, Top2A, and RPA3. At the same time, genes negatively regulated by R-loops that are also H3K36me3 and γH2AX positive include JAK2 and TRIM 14. The association of γH2AX and R-loops with DNA damage repair genes may be important but the mechanism responsible is unclear. Recent studies have suggested that γH2AX might not only interact with sites of endogenous DNA breaks but also associate with DNA intermediates that form upon chromatin opening during transcription initiation [72,73]. The increased expression of genes linked with the combination of γH2AX, R-loops, and H3K36me3 in HPV positive cells compared to normal keratinocytes supports this model.

These studies identify a potential linkage between R-loops, specific histone marks, and altered transcription. However, additional factors must act to determine how genes in specific pathways are targeted. One such possibility may be the association with other non-β DNA structures like G-quadruplexes and GC skew. The relationship between G-quadruplex formation and stability of R-loops has been noted in multiple reports, and may contribute to effects in HPV positive cells [11,74,75]. Similarly, a GC skew has been reported in a number of R-loops, and a preliminary screening indicates that some but not all R-loops associated with innate immune genes have this skew, identifying an important area for future studies. In addition to structural motifs in DNAs, we have shown that inhibition of p53 leads to increased levels of R-loops in HPV positive cells, cells and has been reported in other tumor cell lines that have mutated p53 [20,69]. This indicates that factors downstream of p53 play important roles in regulating R-loop formation and that this occurs at specific sites on cellular genes. Transient inhibition of p53 in normal keratinocytes alone is, however, not sufficient to induce increased R-loop formation but our studies have shown the requirement of HPV E7 co-expression, which implicates inhibition of Rb proteins as a possible contributing factor. Additional factors that could be downstream mediators of the p53 effects on R-loop formation include members of the p21-DREAM complex, long non-coding RNAs, and APOBEC 3B proteins. In embryonic mouse stem cells, a subset of polycomb group genes was shown to be linked with R-loop formation, and overexpression of RNase H1 increased their expression, indicating a repressive effect of R-loops [76]. At the same time, RNase H1 overexpression led to decreased expression of other polycomb genes and this differential regulation is similar to our results. This R-loop dependent activity requires the cooperative action of cellular proteins, and we believe that additional factors, including modified chromatin as well as transcription factors, can provide comparable functions in HPV positive cells. It is also possible that viral proteins can contribute to regulating the expression of R-loop associated genes. Overexpression of RNAse H1 in HPV-positive cells decreased the expression of viral early genes [20], and this reduction in viral proteins could potentially impact the expression of cellular genes that are linked to the presence of R-loops at these loci.

Overall, these observations demonstrate that R-loop levels are significantly elevated within HPV positive cells compared to normal keratinocytes. While no global effect on gene expression is seen due to increased levels of R-loops, genes in pathways that are important for viral replication and cellular transformation are coordinately activated or repressed by these structures, possibly in cooperation with the recruitment of specific types of modified histones. Our studies indicate that in HPV-positive cells, R-loops contribute to regulating cellular and viral gene expression during HPV pathogenesis, including those involved in the innate immune response and DNA damage repair.

## Materials and methods

### Reagents

Antibodies used in these experiments were as follows: S9.6 (Millipore), Anti-Histone H3 (tri methyl K36) antibody—ChIP Grade (Abcam), Anti-Histone H3 (tri methyl K9) antibody—ChIP Grade (Abcam), Phospho-Histone H2A.X (Ser139) (D7T2V) Mouse mAb (Cell Signaling), and Mouse IgG (Diagenode). Methlyene Blue Hydrate (Sigma) was used for staining nucleic acids in dot blot assays. RNase H (ThermoFisher) was used to remove R-loops from nucleic acid extracts to determine specificity of the S9.6 antibody. Mung Bean Nuclease was purchased from New England Biologicals and was used for enzymatic digestion of samples during chromatin immunoprecipitation- and DNA:RNA immunoprecipitation-sequencing.

### Cell culture and reagents

**Isolation of HFKs.** Neonatal human epidermis was supplied by the Skin Disease and Research Core at Northwestern University. These de-identified tissues were suspended in Hanks' balanced salt solution (HBSS), and isolations were performed within 3 to 4 days of circumcision. The foreskins were washed in phosphate-buffered saline (PBS) before being processed. Excess blood vessels, tissue, and fat were cleaned away before being incubated overnight at 4 C in 2.4 U/ml Dispase. The following day, the epidermis was removed and incubated with 4 ml of 0.25% trypsin for 15 min. The epidermis was then scraped vigorously for 2 to 3 min before quenching the trypsin with bovine serum. The resulting suspension was then pipetted through a 40mm pore cell sieve. The cells were then spun down and resuspended in E-medium supplemented with 5ng/ml of mouse epidermal growth factor (EGF). NIH 3T3-J2 fibroblasts, growth-arrested through treatment with mitomycin-c, were seeded with the newly collected human foreskin keratinocytes (HFKs), and media was changed as required until the proliferation of the keratinocytes was achieved.

**Cell culture.** HFKs, HFK-31, and CIN 612 cells were all cultured in E-medium supplemented with 5ng/ml of mouse EGF. Each of these cell lines were co-cultured with NIH 3T3-J2 fibroblasts, which were growth arrested using 0.4mg/ml of mitomycin-c for at least 2 hr. To remove J2 fibroblasts prior to downstream analyses, cells were washed with Versene (0.05mM EDTA PBS) for 5 min before 2 sequential PBS washes. J2 feeders were cultured in DMEM containing 1% penicillin-streptomycin and 10% bovine serum. Cells stably overexpressing RNase H1 were generated previously [20].

**Generation of cells stably maintaining HPV 31 episomes.** The pBR-322min-HPV31 plasmid was digested such that the pBR-322 backbone was removed, leaving the HPV 31 genome which was recircularized. 1 μg of recircularized HPV 31 DNA was contransfected with a selection plasmid expressing a neomycin resistance cassette (PSV2neo) into around one million freshly isolated HFKs at ~60% confluence. The following day, cells were selected using 200 mg/ml G418. J2 feeders were changed on alternating days as the G418 selection. Stable

maintenance of HPV 31 episomal DNA was assessed by Southern blot before expanding and performing downstream analyses on these cells.

### S9.6 dot blot assay

DNA was purified from cell lysates using PhenolChloroform extractions. Samples were either left untreated or treated with 1U of RNase H for at least 1.5 hr at 37˚ C. DNA was then spotted onto a positively charged membrane (Zeta-probe). Membranes were then stained with Methylene blue for ~15 min before being washed with di-H2O 3 times for 5 min. Images of the Methylene blue staining were acquired to normalize to total nucleic acid content using an Odyssey Fc LiCor (LiCor BioSciences). Methylene blue staining was removed through washing with 100% ethanol for 5 min before washing with di-H2O 3 times for 10 min. Membranes were then blocked with 5% Bovine Serum Albumin (BSA) in TBST (Tris-buffered saline Tween 20) before being probed with the S9.6 anti-RNA:DNA hybrid antibody (Millipore) overnight at 4 ˚C. The following day, membranes were washed with TBST, probed with secondary antibody for 1 h at RT, and developed using enhanced chemiluminescence (ECL) (Fisher, 4500085). Images were taken using an Odyssey Fc LiCor.

### DNA:RNA immunoprecipitation (DRIP)–qPCR

$1 \times 10^7$ cells were harvested and collected in Southern lysis buffer before being treated with RNase A (5 ng/mL) and Proteinase K (7.5 ng/mL) at 37 ˚C overnight. DNA was purified from these samples using phenol-chloroform extractions, and 25 to 50 mg of DNA was used for each sample. DNA was sheared using a Bioruptor (Diagenode) on high power, 30 s on/90 s off cycles for 20 min or digested using 1U of mung bean nuclease for 1 h at 37 ˚C. Input DNA was removed before loading the samples into preblocked magnetic beads in IP buffer containing 2 mg of the RNA:DNA hybrid antibody. Immunoprecipitations were allowed to incubate overnight at 4 ˚C while rotating. The next day, samples were washed 8 times with RIPA buffer for 5 min while rotating. One wash in TE buffer was performed before samples were eluted for 10 min at 65 ˚C in 10% sodium dodecyl sulfate (SDS), 10 mM Tris pH 7.4, 50 mM ethylenediaminetetraacetic acid (EDTA). DNA was purified from these elutions using a PCR purification kit (Qiagen) and stored at −20 ˚C. Primer sets used to analyze S9.6 immunoprecipitated sequences are listed in the Key Resources table (Table 1).

### DRIP-sequencing

The same protocol was used to prepare samples for DRIP-sequencing as listed above for DRIP-qPCR. Samples were stored at -80˚ C until being shipped to Admera Biosciences (NJ), who performed the sequencing experiments. Briefly, the library was prepared using a KAPA HyperPrep Kit (Kapa Biosystems) following the manufacturer's recommendation. Input DNA was end-repaired and 3'-dA tailed. Adapter was then ligated to the DNA, and the ligated product was PCR amplified and cleaned up using the SPRIselect Reagent (Beckman Coulter). Quality control was then performed for the final library, followed by sequencing.

### DRIP-sequencing data analysis

Admera Biosciences (NJ) performed most of the bioinformatic analyses from our DRIP-sequencing experiments. Their bioinformatics methods are as follows: An in-house bioinformatics pipeline was used to analyze DRIP-Seq data. First, FastQC (v0.11.8) was used to check the quality of raw and trimmed reads. Trimmomatic (v0.38) was used to cut adapters and trim low-quality bases with a default setting. BWA (v0.7.10-r789) was used to map the

**Table 1. Primer sets used in this study.**

| Primers | | |
|---|---|---|
| **DNA:RNA immunoprecipitation—qPCR primers** | | |
| *Cellular regions* | | |
| MYADM | 5' CGT AGG TGC CCT AGT TGG GAG 3' | 5' TCC ATT CTC ATT CCC AAA CC 3' |
| RPL13a | 5' AAT GTG GCA TTT CCT TCT CG 3' | 5' CCA ATT CGG CCA AGA CTC TA 3' |
| EGR1 | 5' GAA CGT TCA GCC TCG TTC TC 3' | 5' GGA AGG TGG AAG GAA ACA CA 3' |
| SLC35B2 | 5' AAG TCT TGC CCT AGC TGT GCT 3' | 5' GCC TAC ACC GCT TGT GCT TTT 3' |
| SNRPN | 5' GCC AAA TGA GTG AGG ATG GT 3' | 5' TCC TCT CTG CCT GAC TCC AT 3' |
| LGALS2 | 5' TGA CCT CAC CTT GAC CTC TGA 3' | 5' AGC TGA ACC TGC ATT TCA ACC 3' |
| ALU elements | 5' ACG AGG TCA GGA GAT CGA GA 3' | 5' CTC AGC CTC CCA AGT AGC TG 3' |
| *HPV 31 genomic regions* | | |
| Early PolyA | 5' GGT ATT GGT ATT GGT ATT GG 3' | 5' ACC CAT ACT ACC ATA CCT TA 3' |
| Late PolyA | 5' GCG TGT GTA CTT GTA 3' | 5' GCA ACC GAA AAC GGT TAG G 3' |
| Upstream regulatory regions (URR) | 5' GAT GCA GTA GTT CTG CGG TTT 3' | 5' TAT GTT GGC AAG GTG TGT TAG G 3' |

trimmed reads to the reference genome* using the Burrows-Wheeler Alignment algorithm (BWA-MEM). Mapped reads that have low-quality MAPQ score (MAPQ < 10), not-properly-paired, or duplicated (assessed with Picard tools (v 2.20.4)) were removed. BAM was used to generate BW format (normalized by RPKM) for visualization. MACS (v2.2.4) was chosen to call peaks. If there was no replicate, the R package MAnorm (v2.2.6) was used for sample comparison. On the other hand, if there were replicates, their called peaks were merged and the DiffBind package (v2.14.0) was then used for differential analysis. Peak annotation and combined density profiles were performed by the ChIPseeker package (v1.22.1) and deepTools, respectively.

We performed the profile analysis of multiple DRIP-seq replicates from HFKs and CIN 612 cells (Fig 1D) and R-loop read distribution (Fig 2C) using the open-source Galaxy servers https://usegalaxy.org/. BAM Compare (Galaxy Version 3.5.4+galaxy0) was used to normalize either $\log_2$ IP to input ratios, or input subtracted IP reads for both biological replicates of DRIP-seq in HFK and CIN 612 cells. ComputeMatrix was used to prepare files for visualization via plotProfile and plotHeatmap (Fig 1D). CHIPseeker (Galaxy Version 1.28.3+galaxy0) was used on the BED files generated by Admera Biosciences to determine the genomic distribution of R-loop peaks. To confirm agreement between our biological replicates (S7 Fig), correlations between replicates were assessed using multiBAMSummary (Galaxy Version 3.5.4+galaxy0), and then plotting principal component analyses using plotPCA (Galaxy Version 3.5.4+galaxy0) and plotting Pearson coefficients as a heatmap using plotCorrelation (Galaxy Version 3.5.4+galaxy0). plotFingerprint (Galaxy Version 3.5.4+galaxy0) was used to determine narrow versus broad distributions of S9.6 reads across the genome.

## RNA-sequencing

HFKs and CIN 612 cells were grown to confluency on 10cm dishes before removing J2 fibroblasts. Cells were scraped and centrifuged before being stored at -80˚C before shipping to Admera Biosciences (NJ).

## RNA-sequencing data analysis

FastQC (version v0.11.8) was applied to check the quality of raw reads. Trimmomatic (version v0.38) was applied to cut adaptors and trim low-quality bases with default setting. STAR Aligner version 2.7.1a was used to align the reads. Picard tools (version 2.20.4) was applied to mark duplicates of mapping. The StringTie version 2.0.4 was used to assemble the RNA-Seq alignments into potential transcripts. The featureCounts (version 1.6.0)/HTSeq was used to count mapped reads for genomic features such as genes, exons, promoter, gene bodies, genomic bins and chromosomal locations. The De-Seq2 (version 1.14.1) was used to do the differential analysis. Pathway analyses were performed using Shiny GO http://bioinformatics. sdstate.edu/go/.

## Chromatin immunoprecipitation (ChIP)-sequencing

Formaldehyde was added to $1 \times 10^7$ cells to a final concentration of 1% for 10 min at room temperature. Excess formaldehyde was quenched upon adding 0.125M glycine before then washing samples with PBS. Cells were then incubated in collection buffer (0.1M TrisHCl pH 9.4 and 10mM DTT containing Roche Protease Inhibitor Cocktail) for 10min on ice. Cells were then collected and spun down before being sequentially washed and incubated with NCP1 (10mM EDTA, 0.5mM EGTA, 10mM HEPES pH 6.5, 0.25% Triton X100) and NCP2 (1mM EDTA, 0.5mM EGTA, 10mM HEPES, and 200mM NaCl) before being lysed in 0.5% Empigen BB, 1% SDS, 10mM EDTA, 50mM Tris HCl pH 8.0 containing Roche Protease Inhibitor Cocktail for 30 min on ice. Samples were then sonicated using a Bioruptor (Diagenode) on high power, 30 s on/90 s off cycles for 20 min. After sonication, samples were prepared exactly as described above in the DRIP-qPCR protocol. Samples were stored at -80 C before being sent off for sequencing either by Admera Biosciences (NJ) or the NUseq facility at Northwestern University.

## ChIP-sequencing data analysis

Samples were either analyzed as described above in the DRIP-sequencing analysis section or the NU seq core delivered BAM files. Agreement between biological replicates was assessed using multiBAMSummary (Galaxy Version 3.5.4+galaxy0), and then plotting principal component analyses using plotPCA (Galaxy Version 3.5.4+galaxy0) and plotting Pearson coefficients as a heatmap using plotCorrelation (Galaxy Version 3.5.4+galaxy0) (S8 Fig). From the BAM files, BAM Compare (Galaxy Version 3.5.4+galaxy0) was used to normalize $\log_2$ IP to input ratios for both biological replicates of H3K36me3, H3K9me3, and γH2AX ChIPs from HFK and CIN 612 cells. ComputeMatrix was used to prepare files for visualization via plotProfile and plotHeatmap. CHIPseeker (Galaxy Version 1.28.3+galaxy0) was used on the BED files generated by Admera Biosciences or NUseq to determine the genomic distribution of each modified histone. MACS (v2.2.4) was used to call peaks. HOMER (Galaxy Version 4.11+-galaxy0) was used to annotate where peaks occurred relative to their genomic location (intron, exon, etc.) and the corresponding gene name. Gene lists from HOMER were compared between DRIP, H3K36me3 ChIP, H3K9me3 ChIP, and γH2AX ChIP to obtain the overlap depicted in the Venn Diagrams. RNA counts from RNA sequencing experiments were

compared to genes containing the corresponding mark (R-loops, H3K36me3, γH2AX or H3K9me3) to determine the association between mRNA levels seen in Figs 3D, 5A and 6A.

## Quantification and statistical analysis

GraphPad prism was used for all statistical analyses, and all data are represented as mean +/- standard error (SEM). Two-way ANOVA and two-tailed T-tests were used to calculate p-values. Calculation of the representation factor and the associated probability of Venn diagram overlaps in Figs 5B and 6B were performed using http://nemates.org/MA/progs/overlap_stats.html from the Lund Lab. A genome size of 63,755 (CHESS database, http://ccb.jhu.edu/chess) was used to determine representation factors for the Venn Diagrams in Figs 5B and 6B. The maximum value represented as statistically significant was p = 0.05. Additional details on quantifications like replicates are specifically stated in the figure legends and methods.

## Software and algorithms

GraphPad Prism was used to generate all graphs and statistical analyses of said graphs. Adobe Photoshop and Illustrator were used for the organization and preparation of digital figures. Integrated Genome Browser (BioViz) generated depth graphs of S9.6 coverage in HFK and CIN 612 cells (Fig 1E). Galaxy community servers were used to perform many of the sequencing analyses [77].

## Supporting information

**S1 Fig. Input normalized S9.6 reads of two regions associated with R-loops in HFK and CIN 612 cells.** (A) Depth graphs of S9.6 reads normalized to the corresponding cell line's input control reads. Reads were binned into 300bp regions during quantification using deeptools2 (BAMcompare). Two regions are pictured: Lig4 and CALML5. The red represents CIN 612 cells, while the black represents HFKs (n = 2, mean is shown).
(EPS)

**S2 Fig. R-loop association with RNA at different genomic locations does not demonstrate any preferential increases or decreases in either HFKs or CIN 612 cells.** Fold enrichment of S9.6 reads over input was taken from two independent DRIP-sequencing experiments. Genes that contained R-loops in the coding sequence or the corresponding 2kb flanking regions were then analyzed for their mRNA levels from two independent RNA sequencing experiments. These genes were plotted as average mRNA read counts (y-axis) versus fold enrichment of R-loops over input (x-axis). The line of best fit was calculated using Least Squares Regression (GraphPad Prism), where dashed lines represent the 95% confidence intervals (Q = 1% for detection of outliers, red). The equation for the line of best fit is shown for each graph in the top right corner.
(EPS)

**S3 Fig. Pathway analysis of differentially expressed genes in CIN 612 cells overexpressing (o/e) RNase H1 using the Hallmark MSigDB database.** (A) Cumulative numbers of differentially expressed genes in the CIN 612 cells overexpressing RNase H1 versus parental CIN 612 cells (red = downregulated, green = upregulated). The corresponding pathway analysis used Shiny GO 0.80 (http://bioinformatics.sdstate.edu/go/) and the Hallmark MSigDB database (downregulated = left, upregulated = right). Pathways of particular interest were those upregulated upon overexpression of RNase H1, including those responsible for an interferon alpha/gamma response, IL6 JAK STAT3 signaling, and IL2 STAT5 signaling (right). (B) Flow chart of how R-loop dependent gene expression was determined in CIN 612 cells. Negatively

regulated genes were identified as those with reduced expression in parental CIN 612 cells compared to normal keratinocytes, which then increased in expression upon loss of R-loops through RNase H1 overexpression (left). 542 genes were identified, most of which were involved in immune surveillance and signaling. Positively regulated genes were identified as those with increased expression in parental CIN 612 cells compared to normal keratinocytes, which then decreased in expression upon loss of R-loops upon RNase H1 overexpression (right). 722 genes were identified, many of which were involved in DNA metabolism. Of the 1,264 genes identified as being R-loop regulated in CIN 612 cells, 833 of them contained R-loops only in the CIN 612 cells. These genes were deemed as being functionally regulated by R-loop formation for the analyses performed in Fig 4A.
(EPS)

**S4 Fig. H3K36me3 and H3K9me3 are differentially present on host chromatin within CIN 612 cells compared to HFKs.** (A) Venn diagram of the genomic regions containing H3K36me3 peaks (left) and H3K9me3 peaks (right) (MACS) overlapping between the HPV negative (HFKs) and positive (CIN612) cells (n = 2, a representative image is shown). (B) Fingerprint plot of H3K36me3 and H3K9me3 distribution compared to input control in HFKs and CIN 612 cells. (C) Distribution of H3K36me3 and H3K9me3 reads relative to genomic locations in HFKs and CIN 612 cells. CHIPSEEKER was used to analyze the location of each modified histones' reads within each sample (left and right). Deeptools2 (BAMcompare) was used to input normalize H3K36me3 (middle top) and H3K9me3 (middle bottom). Regions analyzed were set as 500bp, flanking the coding sequence, and the average genic profile was visualized (ComputeMatrix). (D) HOMER was used to identify the location of where H3K36me3 (left) or H3K9me3 (right) peaks occurred within HFK and CIN 612 cells. Intergenic histone marks were filtered out, leaving only histone marks that fell within introns, exons, TES, TSS, 3'UTR, and 5'UTR. Common genes found in both HFKs and CIN 612 cells were also filtered out. Pathway analysis was then performed on the genes to which these histone marks were assigned in the CIN 612 cells or the HFKs using Shiny GO 0.80. The GO biological process database was used for all analyses.
(EPS)

**S5 Fig. Pathway analyses of genes enriched with modified histones (γH2AX, H3K36me3, or H3K9me3) and R-loops in CIN 612 cells.** Genes containing a modified histone mark and R-loops unique to CIN 612 cells (left) or HFKs (right) were analyzed using Shiny GO 0.80 and the GO biological process database. Due to the lack of H3K9me3 or γH2AX and R-loop containing genes in HFKs, no analysis of those genes is depicted.
(EPS)

**S6 Fig. γH2AX is significantly enriched on host chromatin and on genes responsible for important processes during HPV infection in CIN 612 cells.** (A) Distribution of γH2AX reads relative to genomic locations in HFKs and CIN 612 cells. CHIPSEEKER was used to analyze the location of R-loop reads within each sample (left and right). (B) Fingerprint plot of γH2AX distribution compared to input control in HFKs and CIN 612 cells. (C) Deeptools2 (BAMcompare) was used to input normalize γH2AX reads. Regions analyzed were set as 500bp, flanking the coding sequence, and the average genic profile was visualized (ComputeMatrix). (D) Venn diagram of the genomic regions containing γH2AX peaks (right) (MACS) overlapping between HFK and CIN 612 cells (n = 2, a representative image is shown). (E) HOMER was used to identify the location of where γH2AX peaks occurred within HFK (bottom) and CIN 612 cells (top). Intergenic histone marks were filtered out, leaving only histone marks that fell within introns, exons, TES, TSS, 3'UTR, and 5'UTR. Common genes found in

both HFKs and CIN 612 cells were also filtered out. Pathway analysis was then performed on the genes to which these histone marks were assigned in the CIN 612 cells or the HFKs using Shiny GO 0.80. The GO biological process database was used for all analyses.
(EPS)

**S7 Fig. Validation of DRIP-sequencing replicates.** (A) XY correlation plots of S9.6 reads in biological replicates from HFKs (top left) and CIN 612 cells (top right). Pearson's coefficient is labeled on each respective scatter plot. XY correlation plot comparing HFK S9.6 reads to CIN 612 S9.6 reads with Spearman's coefficient labeled (bottom middle). These data support that there is a strong agreement between the S9.6 pulldown replicates in HFK and CIN 612 cells and that there are higher read counts in similar genomic regions in the CIN 612 cells. (B) Heatmap of Pearson coefficient values between input controls and S9.6 pulldown assays in HFK and CIN 612 cells. A strong correlation is seen among the S9.6 pulldown assays, suggesting that a majority of the reads are located in similar genomic regions. (C) Fingerprint plot analysis of input control samples and S9.6 pulldown replicates in HFK and CIN 612 cells. Input DNA reads are broadly distributed across the genome, while S9.6 reads are enriched on a much smaller proportion of DNA. (D) Principal component analysis of input control samples and S9.6 pulldown replicates in HFK and CIN 612 cells. A high degree of clustering is seen between the S9.6 replicates from the HFK and CIN 612 cells.
(EPS)

**S8 Fig. Validation of Modified Histones ChIP-sequencing replicates.** (A) Heatmap of Pearson coefficient values between input controls and H3K36me3 pulldown assays in HFK and CIN 612 cells (left). Principal component analysis of input control samples and H3K36me3 pulldown replicates in HFK and CIN 612 cells (right). (B) Heatmap of Pearson coefficient values between input controls and H3K9me3 pulldown assays in HFK and CIN 612 cells (left). Principal component analysis of input control samples and H3K9me3 pulldown replicates in HFK and CIN 612 cells (right). (C) Heatmap of Pearson coefficient values between input controls and γH2AX pulldown assays in HFK and CIN 612 cells (left). Principal component analysis of input control samples and γH2AX pulldown replicates in HFK and CIN 612 cells (right).
(EPS)

## Acknowledgments

The Skin Biology and Diseases Research Core at Northwestern University provided foreskin keratinocytes.

## Author Contributions

**Conceptualization:** Conor W. Templeton, Laimonis A. Laimins.

**Data curation:** Conor W. Templeton.

**Formal analysis:** Conor W. Templeton, Laimonis A. Laimins.

**Funding acquisition:** Conor W. Templeton, Laimonis A. Laimins.

**Investigation:** Conor W. Templeton, Laimonis A. Laimins.

**Methodology:** Conor W. Templeton, Laimonis A. Laimins.

**Project administration:** Laimonis A. Laimins.

**Resources:** Conor W. Templeton, Laimonis A. Laimins.

**Software:** Conor W. Templeton.

**Supervision:** Conor W. Templeton, Laimonis A. Laimins.

**Validation:** Conor W. Templeton, Laimonis A. Laimins.

**Visualization:** Conor W. Templeton, Laimonis A. Laimins.

**Writing – original draft:** Conor W. Templeton, Laimonis A. Laimins.

**Writing – review & editing:** Conor W. Templeton, Laimonis A. Laimins.

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
