## [Decision Letter · Decision Letter 0]

21 Jun 2024

Dear Dr. Laimins,

Thank you very much for submitting your manuscript "HPV induced R-loop formation represses innate immune gene expression while activating DNA damage repair pathways" for consideration at PLOS Pathogens. As with all papers reviewed by the journal, your manuscript was reviewed by members of the editorial board and by several independent reviewers. In light of the reviews (below this email), we would like to invite the resubmission of a significantly-revised version that takes into account the reviewers' comments.

We cannot make any decision about publication until we have seen the revised manuscript and your response to the reviewers' comments. Your revised manuscript is also likely to be sent to reviewers for further evaluation.

Sincerely,

Paul F. Lambert

Academic Editor

PLOS Pathogens

Robert Kalejta

Section Editor

PLOS Pathogens

Michael Malim

Editor-in-Chief

PLOS Pathogens

orcid.org/0000-0002-7699-2064

Reviewer's Responses to Questions

**Part I - Summary**

Reviewer #1: This report by Templeton and Laimins is the follow-up to their recent PNAS paper in which they initially describe a role for R-loops in HPV infection. Overall, this study fills in some of the remaining gaps of their PNAS study. Using an assortment of NEXTgen sequencing, the authors confirm that R-loops are significantly upregulated in HPV(+) cells and are associated with different genes compared to control HFKs. There appears to be an overlap between differentially expressed genes and R-loop formation. To show that R-loops regulate the differences in expression, the authors overexpress RNase H1 in the CIN612 cells. The authors see the expected decrease in R-loops and also observe a difference in overall gene expression compared to the vector expressing cells. However, in the previous report, the authors show that overexpression of RNAse H1 impacts the viral genome. So it is not clear to me how the authors can distinguish between effects due to R-loop resolution and reduced viral gene expression. I would like to see this addressed by the authors. For example, how does RNAase H1 alter viral gene expression?

In the last part of the manuscript, the authors correlate gene expression, R-loops, and chromatin marks. The authors use venn diagrams to calculate overlap. Previous studies formally compared the R-loop and chromatin mark ChIP-Seq peaks to demonstrate overlap (e.g., 10.1016/j.molcel.2016.05.032). The authors should consider using that approach as a more sensitive and relevant option.

Reviewer #2: In the manuscript entitled “ HPV induced R-loop formation represses innate immune gene expression while activating DNA damage repair pathways”, the authors investigated the levels of co-transcriptional RNA:DNA hybrid (R-loop) structures in normal non-cancerous keratinocytes and pre-cancerous cells containing human papilloma virus (HPV) genomes. The authors discover higher R-loop levels in HPV-positive cells both at coding genes and repetitive sequences. Interestingly, genes associated with higher R-loops showed increased H3K36me3 levels and both positive and negative impact on gene expression, with upregulated genes having a function in DNA damage repair and metabolism, and downregulated genes having a function in innate immuneresponses. Thus, R-loops may provide a critical link in HPV positive cells to allow HPV pathogenesis in the host cells.

Overall, I find this an interesting study that shows clearly differences between HPV infected and non-infected cells and the suggested role of R-loops to differentially up or downregulate certain gene categories and thereby allows the virus to exploit the host genome is an interesting concept that should be reported. However, I have a few major comments that should be addressed prior to publication:

**Part II – Major Issues: Key Experiments Required for Acceptance**

Reviewer #1: 1) RNase H1 overexpression impacts viral maintenance and transcription. This should be considered in the interpretation of the RNAsH1 overexpression experiments.

2) Consider using peak-calling algorithms to calculate overlap between R-loops and chromatin marks.

Reviewer #2: 1) The DRIP-Seq data presented in Figure 1D show 3 major peaks of R-loop accumulation: i) ~1kb upstream of TSS, ii) over the TES and iii) ~1kb downstream of TES. This seems an unusual distribution of co-transcriptional structures that should primarily track with transcriptional activity. Especially peaks about 1kb upstream of TSS is rather unusual and typically not seen in other DRIP-Seq datasets. I think it would be important to further analyze this data and compare it directly with other DRIP-Seq datasets (e.g how the distribution of peaks with promoters, gene bodies, exons, introns, intergenic regions, etc. (Figure 2C) compares with other datasets.

2) The authors conclude from the DRIP-Seq data set that “only a minority of genes exhibited distinct patterns of R-loop formation in CIN 612 compared to normal keratinocytes”. Even if only a minority, this subset of genes would be very interesting to look at in more detail but the authors didn’t further explore this category and only focused on the genes that showed enhanced R-loop formation but at the same position.

3) The DRIP-Seq is a key dataset of the paper and the authors find quantitative but arguably small differences between the CIN612 and HFK cell lines. As I could not find this information, did the authors use a spike-in for normalization between samples? How do the two biological replicates correlate with each other? These would be important quality controls to assure that the differences are biologically significant and relevant.

4) The authors show in Figure 2C that ~ 60% of the DRIP-Seq reads are distributed within genic regions in both cell lines which agrees well with current literature. However, this is not in agreement with the heatmap provided in Figure 1D where it seems that the majority of peaks are upstream and downstream of genic regions. Why?

5) It is very interesting that the genes that increase R-loop levels are associated with both up and downregulated gene expression (Figure 3), which begs the question whether R-loops are cause or consequence for this transcriptional deregulation. For example, the genes could be upregulated and R-loops form as a consequence of this higher transcriptional activity. On the other hand, more stable R-loop formation could impede transcription elongation and therefore be the cause of downregulation of genes. Can the authors further analyze the subset of genes that are either up-or downregulated and see whether these genes have for example specific sequence features such as higher GC content, GC-skew, or potential for forming other secondary structures such as G4s?

6) A surprising result is presented in Figure 5A (right panels) where the authors show that genes associated with H3K9me3 (a marker for heterochromatin) in HFKs is associated with increased mRNA levels. This is not further commented. Can the authors check again their method to call H3K9me3+ versus H3K9me3- regions and make sure the thresholds they use to binarize the data are appropriate?

**Part III – Minor Issues: Editorial and Data Presentation Modifications**

Reviewer #1: 1) Please provide the details on how the venn diagram based overlaps are calculated. Specifically, what is the 'genome' size that is used as background.

2) Venn diagrams rely on arbitrary cut-offs as to what genes to include. See point 2 above about peak calling.

3) The authors demonstrate correlations between R-loops, gene expression, and chromatin marks. They should be careful not to overstate the importance of this. Without more data, it is hard to draw strong conclusions.

Reviewer #2: None

PLOS authors have the option to publish the peer review history of their article (what does this mean?). If published, this will include your full peer review and any attached files.

Reviewer #1: No

Reviewer #2: No
---

## [Decision Letter · Decision Letter 1]

28 Jul 2024

Dear Dr. Laimins,

We are pleased to inform you that your manuscript 'HPV induced R-loop formation represses innate immune gene expression while activating DNA damage repair pathways' has been provisionally accepted for publication in PLOS Pathogens.

Best regards,

Paul F. Lambert

Academic Editor

PLOS Pathogens

Robert Kalejta

Section Editor

PLOS Pathogens

Michael Malim

Editor-in-Chief

PLOS Pathogens

orcid.org/0000-0002-7699-2064

Reviewer Comments (if any, and for reference):

Reviewer's Responses to Questions

**Part I - Summary**

Reviewer #1: Authors have addressed the concerns adequately

Reviewer #2: The authors have done a fair amount of additional analysis and provided sufficient explanations to justify their results and conclusions. I agree that this study can be published in its current form and don't have additional questions or issues.

**Part II – Major Issues: Key Experiments Required for Acceptance**

Reviewer #1: N/A

Reviewer #2: See above

**Part III – Minor Issues: Editorial and Data Presentation Modifications**

Reviewer #1: N/A

Reviewer #2: See above

PLOS authors have the option to publish the peer review history of their article (what does this mean?). If published, this will include your full peer review and any attached files.

Reviewer #1: No

Reviewer #2: No

---

## [Editor Report · Acceptance letter]

14 Aug 2024

Dear Dr. Laimins,

We are delighted to inform you that your manuscript, "HPV induced R-loop formation represses innate immune gene expression while activating DNA damage repair pathways," has been formally accepted for publication in PLOS Pathogens.

Best regards,

Michael Malim

Editor-in-Chief

PLOS Pathogens

orcid.org/0000-0002-7699-2064